# Phytoplankton Dynamics and Biogeochemistry of the Black Sea

**Vladimir Silkin \*, Larisa Pautova, Oleg Podymov** 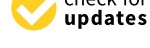**, Valeryi Chasovnikov, Anna Lifanchuk** **, Alexey Fedorov and Agnislava Kluchantseva**

Shirshov Institute of Oceanology, Russian Academy of Sciences, 117977 Moscow, Russia;
larisapautova@ocean.ru (L.P.); huravela@yahoo.com (O.P.); chasovn@mail.ru (V.C.);
lifanchuk.anna@mail.ru (A.L.); aleksey_fedorov_199001@mail.ru (A.F.); agni_text@mail.ru (A.K.)
\* Correspondence: vsilkin@mail.ru; Tel.: +7-918-200-7560

**Abstract:** The biogeochemistry of waters is an essential regulator of phytoplankton dynamics, determining the level of species bloom and the change in dominants. This paper investigated the seasonal dynamics of phytoplankton and the nutrient concentrations and their ratios in the northeastern Black Sea in 2017–2021. Two taxonomic groups, diatoms and coccolithophores, determine the seasonal dynamics and significantly contribute to the total phytoplankton biomass. Coccolithophores formed blooms in early June annually, except in 2020. Large diatoms dominated in summer with a biomass exceeding 1000 mg m$^{-3}$ annually, except in 2019. During the blooms of these taxonomic groups, their contribution to the total phytoplankton biomass exceeded 90%. Each group has characteristic biogeochemical niches in the nitrogen and phosphorus concentration coordinates. The position of the seasonal thermocline regulates the biogeochemistry of the water. With a high-lying and sharp gradient thermocline (the average for five years is 6.87 m), low nitrogen concentrations and a nitrogen-to-phosphorus ratio below the Redfield ratio are created in the upper mixed layer. These conditions are optimal for the dominance of coccolithophores. When the thermocline is deepened (the average for five years is 17.96 m), the phosphorus concentration decreases significantly and the ratio of nitrogen to phosphorus is significantly higher than the Redfield ratio, and these conditions lead to the dominance of large diatoms. The results of experimental studies with nitrogen and phosphorus additives in the natural phytoplankton population confirm the above statements. The addition of phosphorus leads to the increased role of coccolithophores in the total phytoplankton biomass, the addition of nitrogen alone contributes to the growth of large diatoms, and the combined addition of phosphorus and nitrogen in a ratio close to the Redfield ratio leads to the dominance of small diatoms.

**Keywords:** phytoplankton; Black Sea; biogeochemistry; species composition; diatoms; coccolithophores

## 1. Introduction

The phytoplankton of the World Ocean plays an essential role in regulating the planet's climate, and it accounts for about half of the global primary productivity, i.e., carbon dioxide assimilation [1]. Atmospheric carbon is transported to the deep layers of the ocean due to the functioning of the biological carbon pump (BCP) [2,3]; existing estimates of global carbon exports are in the range from 5 to $15 \times 10^{15}$ g C per year [4,5]. The BCP has a complex structure and includes organic and carbonate pumps [6]. Various taxonomic and functional groups are responsible for the latter's work [7]. The organic pump is mainly represented by diatoms [8], the carbonate pump includes calcifying organisms, and in the ocean, more than 50% of the work of the carbonate pump consists of coccolithophorids capable of building a cell shell from $CaCO_3$ [9,10]. In the process of calcification, $CO_2$ is released into the seawater, which reduces the ability of the ocean to absorb atmospheric carbon. Therefore, the ratio of diatoms and coccolithophorids is fundamental for regulating the planet's climate [7]. In this regard, the question arises of what factors influence this

ratio. First, the question is raised about the contribution of the biogeochemical regime of water masses to the regulatory role of this ratio.

The ratio of carbon, nitrogen, and phosphorus (C:N:P) discovered by Redfield, i.e., an atomic ratio of C106:N16:P1 in the particulate organic matter (POM), has become a kind of reference point in ocean biogeochemistry, with which large-scale biogeochemical processes are considered [11].

This ratio has been a paradigm for a long time and has been widely used in biogeochemical models [12]. It was later shown that C:N:P has spatial variability [11,13]. In warm near-equatorial waters, C:N:P is higher than the Redfield ratio; when moving to the poles, it decreases, and at high latitudes, it is significantly lower than the canonical ratio. C:N:P is characterized by temporal variability on the scale of days, seasons, and years [11].

It has been shown that there are two main factors of plasticity in C:N:P—physiological and taxonomic [11,14]. At the level of one species, this ratio is a function of the concentration of these substances in the medium, and this is a physiological response. On the other hand, the different ratios of these elements are characteristic of different species, which is a taxonomic answer. Furthermore, the latitudinal variability of N:P is due to a taxonomic difference: in southern latitudes, small-size phytoplankton (mainly prokaryotes) with a high value of this indicator dominate; in high latitudes, large eukaryotes contribute to a decrease in N:P [15,16].

In ecological stoichiometry, the dominant hypothesis is the growth rate hypothesis, which postulates that the growth rate positively correlates with the macromolecular composition of biomass and, above all, with the content of ribosomes. Consequently, in fast-growing species, the ribosome content should be higher, which leads to the fact that N:P should be lower [17–19]. This pattern has been shown in different taxa and across a broad size spectrum [14,19–22]. Phytoplankton are no exception [23]. It follows that the potential production properties can be judged based on the biochemical composition of the biomass, which is taxonomically determined [24,25]. N:P depends on the cell's ratio of ribosomes and protein [26–28], and therefore, species with a high growth rate will have a low cell N:P ratio [29].

Applying the growth rate hypothesis to phytoplankton has several features [30]. In particular, in nitrogen-limiting growth media N:P may not increase with an increasing growth rate. The maximum specific growth rate of the species is noted in the exponential growth phase, and here will be found the optimal ratio of elements for maximum growth. In addition, polyphosphates can serve as the storage reservoir of phosphorus in the cell [31], which significantly changes the stoichiometry.

It is fundamentally essential that the ratio of elements in the biomass is a function of their concentration in the medium, and the content of the growth-limiting element and the specific growth rate are related. The Droop equation is a reasonably simple equation describing this relationship [32,33]. This equation has gained the most significant prevalence for creating and improving phytoplankton growth models [34–36].

Currently, attempts are being made to link ecological stoichiometry with traits of fundamental importance for growth, survival, and competition [37]. These features are associated with the size of phytoplankton cells using allometric dependencies [38–43].

The Redfield ratio is widely used to compare concentrations in water, primarily nitrogen and phosphorus, in various ocean biomes and to identify a factor limiting the productivity of phytoplankton. It is assumed that nitrogen limits phytoplankton growth when N:P < 16:1; when N:P > 16:1, phosphorus becomes the limiting factor [44]. It has been shown that N:P in the phytoplankton biomass is close to the ratio of these elements in the ocean. Two main points of view explain this fact [45]. The first hypothesis is that modern phytoplankton result from the long evolution of phytoplankton in historically existing conditions when modern phytoplankton taxonomic groups were formed. The second concept argues that the concentration of nutrients and their ratio determine by the phytoplankton, and this hypothesis is currently dominant [26]. The composition of phytoplankton is determined by the ratio of nitrogen and phosphorus in the incoming

water [46,47]. The residual nutrient concentration and their ratio in the water depend on phytoplankton species [28,48]. The spatial and temporal variability of the biogeochemistry of the waters of the World Ocean indicates the need to investigate these features of the research area. In our work, research was carried out in the Black Sea, which has some unique properties.

The Black Sea is an inland sea with a relatively weak influence from the ocean; the salinity here is half the oceanic [49]. This sea location affects its hydrophysical and hydrochemical properties [50]. The biogeochemistry of the Black Sea water masses is subject to significant spatial and temporal variations [51–54]. The rivers' discharge determines the biogeochemistry of this sea. The main river flow is 80% located in the western part of the sea, and this part of the sea is more eutrophicated [55–58]; in the eastern part of the sea, eutrophication is less and such spatial variability affects the state of the marine ecosystem [59]. A unique feature of the Black Sea is a Rim Current that runs along the continental slope and forms a cyclonic circulation [60,61]. The Black Sea has a complex vertical hydrophysical structure characterized by a cold intermediate layer and a pycnocline separating the waters of the Black Sea proper from the underlying Mediterranean waters [62]. The vertical biogeochemical structure is determined by oxic, suboxic, and anoxic layers [62]. The vertical nutrient flow significantly depends on the weather conditions, in particular on the winter temperatures of the atmosphere [63] and the thickness of the upper mixed layer [64].

The phytoplankton composition is determined by two taxonomic groups: diatoms and coccolithophores [65–68]. Coccolithophore blooms occur almost annually in late May and early June and occupy large areas [69,70]. In the northeastern part of the Black Sea, studies from 2002 to 2016 showed that the biogeochemistry of the water determines the annual dynamics of phytoplankton [68]. The development of diatoms and coccolithophores is associated with the concentrations of silicon, nitrogen, and phosphorus and their ratios. However, climate change and variations in the anthropogenic load may be the reason for the shifts in the biogeochemical functioning of the Black Sea ecosystem. Therefore, assessing the current biogeochemical status is an important stage for understanding the processes that lead to changes in the ecosystem.

In this paper, the following hypothesis is tested: the blooms of coccolithophores and large diatoms determine the functioning of the ecosystem of the northeastern part of the Black Sea at the current; these phenomena depend on the biogeochemistry of the water and, first of all, on the concentration of nitrogen and phosphorus and their ratio. For this purpose, the data of field observations for the last five years (2017–2021) have been analyzed and supplemented with the results of experimental studies of the effect of nitrogen and phosphorus additives on the phytoplankton composition.

## 2. Methods

### 2.1. Field Sampling

Physical and chemical data and phytoplankton composition studies were carried out on numerous cruises of R/V Ashamba from 2017 to 2021 in the northeastern part of the Black Sea (Figure 1). In 2017, 8 cruises were conducted (from 28 April to 4 September); in 2018, 6 cruises (5 June–9 October); in 2019, 12 cruises (8 April–26 November); in 2020, 15 cruises (11 June–27 November); and in 2021, 3 cruises (7.June–17 August). The sample stations were located on a transect from Golubaya Bay (the region of Gelendgik City). The stations were located at depths of 25, 50, 100, 500, 1000, and 1500 m. A CTD (Sea-Bird Electronics, Inc., Washington, DC, USA) was used to measure temperature, salinity, and density and thence to estimate the upper mixed layer (UML) depth.

Niskin bottles of 5 L mounted on a Rosette sampler were used for water sampling. At each station, samples were taken from different depths (the water surface, the middle of the upper mixed layer for phytoplankton composition (UML), the seasonal thermocline, and below the thermocline). The samples (bottles of 1–1.5 L) were fixed with neutralized formaldehyde (0.8–1.0% final concentration). Samples were stored in darkness, at room

temperature (18–20 °C), for two weeks; after cell sedimentation, the upper water layers were slowly decanted.

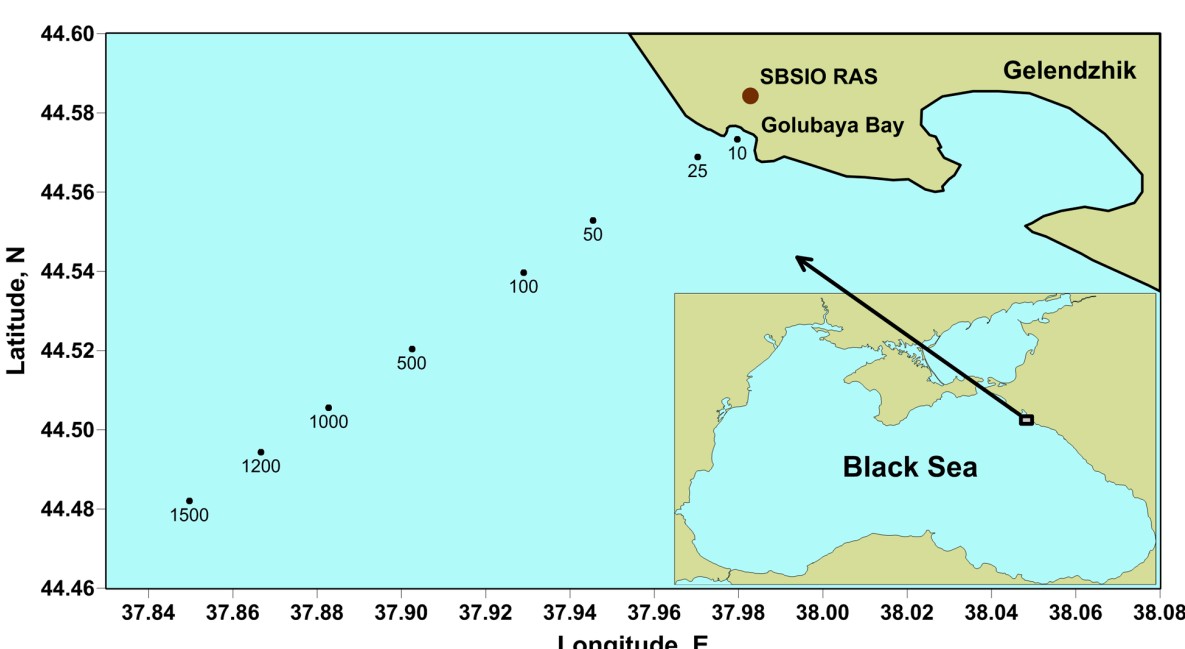

**Figure 1.** Map of sampling stations in the northeastern part of the Black Sea 2017–2021.

## 2.2. Field Sample Analyses

Standard spectrophotometric techniques were used to measure the concentrations of silicate, phosphate ($PO_4^-$), nitrate ($NO_3^-$), nitrite ($NO_2^-$), and ammonium ($NH_4^+$) [71,72]. Total dissolved inorganic nitrogen (DIN) was estimated as the sum of nitrate, nitrite, and ammonium.

Light microscopy (Jenaval, Carl Zeiss "Jena", Jena, Germany) with $16 \times 20$ and $16 \times 40$ magnifications was used for species identification and cell counting. Cells with linear dimensions of less than 20 µm were counted with a Naujotte chamber (0.05 mL), and large cells were counted with a Naumann (1 mL) chamber.

Species identification was based on morphology guided by the following sources: Tomas (1997) and Throndsen et al. (2003) [73,74], and the World Register of Marine Species (http://www.marinespecies.org, accessed on 1 December 2022). Cells with unknown taxonomic affiliation with linear dimensions of 4 to 10 µm were accepted as "small flagellates." Cell volume was estimated according to Hillebrand et al. (1999) [75]. Wet weight was calculated from the cell biovolume, assuming cell density equals 1 g mL$^{-1}$. The standard equations for converting biovolume to carbon biomass units were used [76]. The dominant species was considered as the species with maximal biomass at this station. The subdominant species was the species with the second greatest biomass.

## 2.3. Experimental Studies

The experimental study of nutrient treatment on phytoplankton dynamics in a natural population was performed at the coastal laboratory of the Southern Branch of the P.P. Shirshov Institute of Oceanology (Gelendzhik). Water for the experiments was filtered through two layers of the net (mesh size—180 µm) to remove zooplankton and was transferred to a 0.5-L Erlenmeyer flask. The volume of the culture medium was 200 mL in all experiments. These flasks were kept in a growth chamber where the temperature and irradiance were regulated. Water temperature was maintained at the same value as the sea surface at the time of sampling. Lighting was supplied using LEDs (SMD 5050, cold white light, 6500 K), and the photon flux density (PFD) was 50–60 µmol photons m$^{-2}$ s$^{-1}$ PAR; the light–dark cycle was 16:8. Enrichment treatments were realized by the nitrogen and

phosphorus supply to the final concentration 12–14 µM and 1 µM, respectively. The scheme of full factorial design 22 was used for the experiment [77]. Four variants of experiments were used in this scheme (Table S1): 1—without nutrients addition; 2—only the addition of nitrogen; 3—only the addition of phosphorus; 4—nitrogen and phosphorus additions simultaneously. In each variant, the temperature and irradiance were constant and did not change throughout the experiment. Each variant was assessed in duplicate or triplicate; the Student's *t*-test was used to compare different variants in the experiments, and the significance was set at 5%. A batch method of cultivation was applied in all experiments. Counts of the cell abundances were carried out every day. Algae of all systematic and size groups were considered, except for the picoplankton fraction (0.2–2 µm).

## 3. Results

### *3.1. Dominance of Coccolithophores and Large Diatoms in the Black Sea*

#### 3.1.1. Dominance of Coccolithophores

The dominance of coccolithophores usually occurs in early June every year. Only one species, *Emiliania huxleyi*, represented coccolithophores. The vertical distribution was characterized by the fact that the bloom of this species was observed only in the upper mixed layer (UML) from the surface to 15 m (Figure 2). The bloom level was reached in 2017, 2018, 2019, and 2021, and the maximum cell abundance was observed in 2017 ($9.6 \times 10^6$ cells $L^{-1}$) (Figure 3, Table S2). At the same time, the coccolithophores' contribution to the total phytoplankton biomass could reach 99% (Table S2). The biomass of diatoms during the coccolithophores bloom was always negligible. The exception was one case registered at an offshore station when, on 11.06. 2017, at a depth of 9 m, an accumulation of *Cerataulina pelagica* with biomass exceeding 1000 mg m$^{-3}$ was detected. The biomass of dinoflagellates was always low and did not reach 100 mg m$^{-3}$, except for one case on 5.06. 2019, when an accumulation of *Alexandrium ostenfeldii* with biomass about 2000 mg m$^{-3}$ was recorded on the water surface of the shelf station above a depth of 25 m. The contribution of small flagellates to the total phytoplankton biomass was always low.

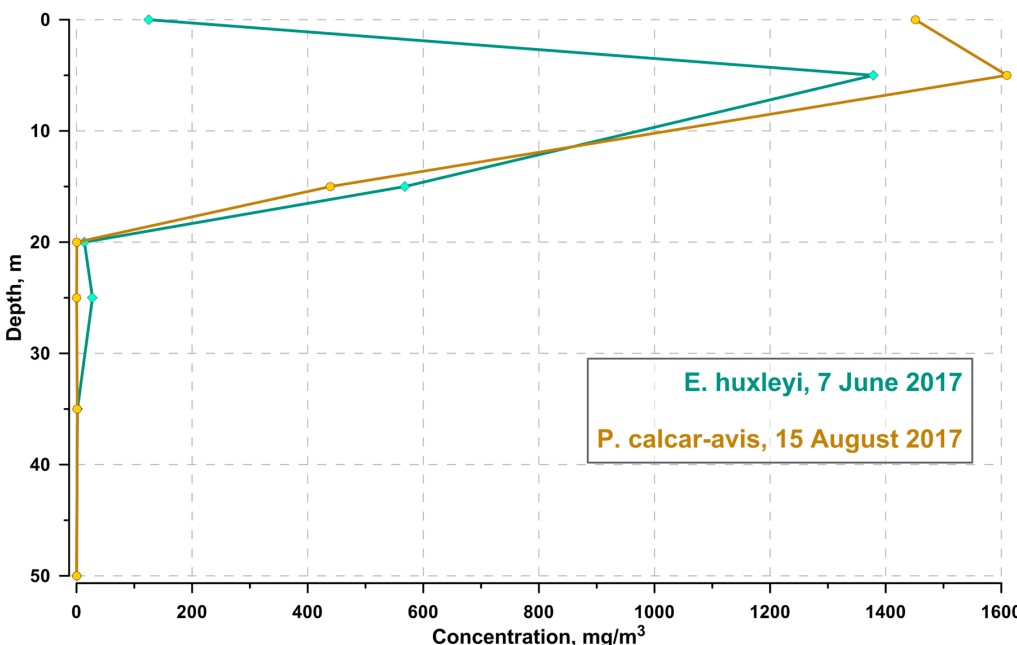

**Figure 2.** Vertical distribution of biomass concentrations of *Emiliania huxleyi* in 7 June 2017 and *Pseudosolenia calcar-avis* in 15 August 2017.

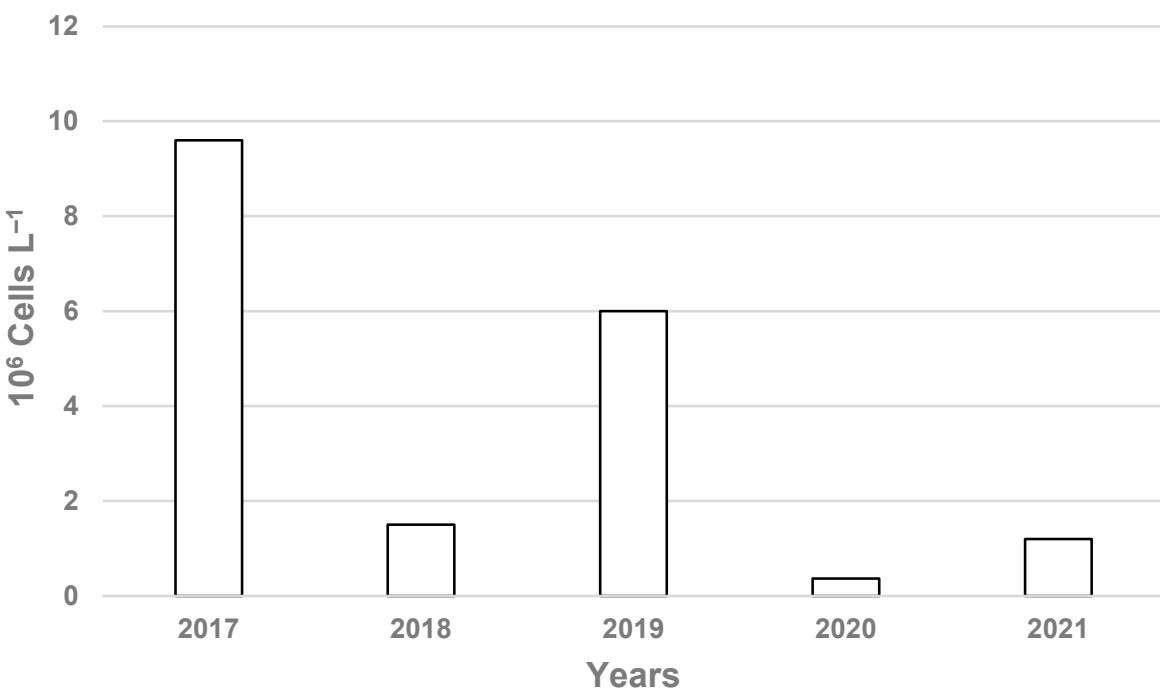

**Figure 3.** Dynamics of maximum values of *Emiliania huxleyi* abundance at research stations in 2017–2021.

As for the spatial distribution, studies conducted in 2017 and 2019 showed that blooms covered the entire study area (Table 1). No significant differences existed in *Emiliania huxleyi* abundance between shelf and slope stations.

**Table 1.** *Emiliania huxleyi* abundance ($\times 10^6$ cells L$^{-1}$) on the shelf and slope in June 2017 and 2019.

| Shelf | Slope | Shelf | Slope | Shelf | Slope | Shelf | Slope |
|-------|-------|-------|-------|-------|-------|-------|-------|
| **8 June 2017** | | **9 June 2017** | | **10 June 2017** | | | |
| 5.96 | 4.75 | 7.06 | 7.5 | 7.22 | 6.4 | | |
| *p* = 0.54 | | *p* = 0.69 | | *p* = 0.26 | | | |
| **5 June 2019** | | **6 June 2019** | | **7 June 2019** | | **8 June 2019** | |
| 2.51 | 3.04 | 3.69 | 3.02 | 2.2 | 2.39 | 3.42 | 2.76 |
| *p* = 0.375 | | *p* = 0.54 | | *p* = 0.69 | | *p* = 0.45 | |

### 3.1.2. The Dominance of Large Diatoms

The dominance of large diatoms in July–August 2017–2021 was recorded annually; maximum biomass occurred in the UML from the surface to a depth of 10–18 m (Figure 2, Table 2). At the same time, the maximum biomass of large diatoms always exceeded 1000 mg m$^{-3}$, with the exception of 2019. The biomass of large diatoms was due to the intensive growth of almost one species: *Pseudosolenia calcar-avis*. The maximum biomass of this species was recorded in July 2020 at a depth of 15 m and in August 2021 at the surface. In addition to this species, in 2020 and 2021, another large diatom *Proboscia alata* made a significant contribution to the biomass of large diatoms. (Tables 2 and 3).

**Table 2.** Maximum biomass of large diatoms, their contribution to the total phytoplankton biomass, and dominant species in the summers of 2017–2021.

| Date | Max Biomass | % of Total Biomass | Dominant Species |
|---|---|---|---|
| July 2017 | 1609.75 | 98.9 | *Pseudosolenia calcar-avis* |
| July 2018 | 1121.0 | 91.0 | *Pseudosolenia calcar-avis* |
| July 2019 | 214.0 | 67.3 | *Pseudosolenia calcar-avis* |
| July 2020 | 4300,9 | 98.5 | *Pseudosolenia calcar-avis + Proboscia alata* |
| August 2021 | 3914.0 | 95.3 | *Pseudosolenia calcar-avis + Proboscia alata* |

**Table 3.** Biomass of large diatoms in the summers of 2017, 2020, and 2021.

| Species | 17 August 2017 | 22 July 2020 | 31 July 2020 | 16 August 2020 | 26 August 2020 | 17 August 2021 |
|---|---|---|---|---|---|---|
| *Proboscia alata* | 0 | 112.9 | 902.6 | 557.6 | 5.4 | 1298.5 |
| *Pseudosolenia calcar-avis* | 1430.4 | 3496.2 | 556.7 | 124.0 | 356.8 | 2058.4 |
| *Dactyliosolen fragilissimus* | 0 | 0 | 0 | 0 | 0 | 500 |
| *Pseudo-nitzschia delicatissima* | 13.6 | 0 | 0 | 0 | 0 | 0 |

At the end of July and in the first half of August 2020, this species was the dominant species. In August 2021, *Proboscia alata* was the subdominant species, but its biomass significantly exceeded 1000 mg m$^{-3}$. Also, at this time, another large diatom, *Dactyliosolen fragilissimus*, with relatively high biomass, was recorded. In August 2017, the pennate small-cell diatom *Pseudo-nitzschia delicatissima* appeared for a short time in the phytoplankton.

During the bloom of large diatoms, the contribution of dinoflagellates, coccolithophorids, and small flagellates was always relatively low (Table S3).

*3.2. Physics of the Water Column*

The physics of the water column depended on the absorption of the heat flux generated by solar radiation (Figure S1). The maximum light flux were recorded in June and July, and the minimum in December and January; the difference was about five times. The northeastern part of the Black Sea is non-freezing; observations over the past 20 years show that the minimum water temperature on the surface is in February and March and is in the region of 9–10 °C (Figure S2). The maximum temperature reaches 28 °C in August.

The bloom of the coccolithophore *Emiliania huxleyi* occurred in late spring and early summer in the UML with a high-lying thermocline (Table 4, Figure S3). Data for 2017–2021 showed that the water temperature in the UML during the bloom of the coccolithophore *Emiliania huxleyi* approached 21 °C. The depth of the water layer of 10 °C, which corresponded to the winter temperature on the water surface, was low. Accordingly, the thickness of the layer from the thermocline to the reference temperature of 10 °C was also relatively small.

The temperature and salinity in the UML and the depth and thickness of the seasonal thermocline during large diatom *Pseudosolenia calcar-avis* blooms in summer were observed with a deep-lying thermocline and a water temperature in the UML reaching 26 °C (Table 1). The water layer with a temperature of 10 °C was located much lower than in late spring and early summer during the coccolithophore bloom. The thickness of the thermocline also increased. In summer, there was an increase in salinity on the surface water.

**Table 4.** Temperature and salinity at UML; depth and thickness of seasonal thermocline during *Emiliania huxleyi* and large diatom *Pseudosolenia calcar-avis* blooms.

| Parameter | E. huxleyi | P. calcar-avis | p, t-Test |
|---|---|---|---|
| Temperature | 20.89 | 25.93 | $5.1 \times 10^{-26}$ |
| Salinity | 17.74 | 17.97 | 0.0004 |
| Depth of seasonal thermocline | 6.87 | 17.96 | $1 \times 10^{-7}$ |
| Depth of layer 10 °C | 27.47 | 42.67 | $5.1 \times 10^{-7}$ |
| Thickness of seasonal thermocline | 20.60 | 24.41 | 0.031 |

### 3.3. UML Chemistry

When the coccolithophores *Emiliania huxleyi* dominated in early summer, low nitrogen concentrations were recorded, but relatively high phosphorus concentrations and N:P were below the Redfield ratio (Table 5). At this time, high silicon concentrations were observed, which led to an increase in Si:N.

**Table 5.** Average concentrations of nitrogen, phosphorus, and silicon and their ratios during coccolithophorid *Emiliania huxleyi (E.h)* and large diatom *Pseudosolenia calcar-avis (P.c-a)* blooms.

| N | | P | | Si. | | N:P | | Si:N | | Si:P | |
|---|---|---|---|---|---|---|---|---|---|---|---|
| E. h. | P. c-a | E. h. | P. c-a | E. h. | P. c-a | E. h. | P. c-a | E. h. | P. c-a | E. h. | P. c-a |
| 44 | 86 | 44 | 86 | 44 | 86 | 44 | 86 | 44 | 86 | 44 | 86 |
| 0.77 | 1.04 | 0.10 | 0.04 | 5.54 | 2.19 | 12.63 | 60.29 | 8,74 | 4,35 | 99.4 | 87.0 |
| $p = 0.04$ | | $p = 1.7 \times 10^{-8}$ | | $p = 6.25 \times 10^{-13}$ | | $p = 0.0003$ | | $p = 0.003$ | | $p = 0.51$ | |

The nitrogen concentration significantly increased during the bloom of the large diatoms *Pseudosolenia calcar-avis*. However, the phosphorus concentration significantly decreased, as a result of which the N:P became significantly higher than the Redfield ratio (Table 4). In addition, the silicon concentration was significantly reduced, which led to a significant decrease in Si:N. During the *Emiliania huxleyi* bloom and the large diatoms *P. calcar-avis* bloom, the Si:P ratios were statistically indistinguishable.

### 3.4. The Effect of Nitrogen and Phosphorus Additives on Phytoplankton Dynamics

In an experiment conducted on 22 May 2019, an intensive growth of *Emiliania huxleyi* was observed in a variant with the simultaneous addition of nitrogen and phosphorus (Table 6). The regression equations showed that phosphorus's contribution exceeded nitrogen's contribution (Table 7). At the same time, the growth of the small pennate diatom *Pseudo-nitzschia delicatissima* was observed in the population; it was maximal with the simultaneous addition of nitrogen and phosphorus, while the contribution of nitrogen was one and a half times higher than the contribution of phosphorus. The growth of a large diatom was noticeably higher in the absence of phosphorus supplementation. In an experiment conducted on 11 June 2020, the growth of *E. huxleyi* was observed only in experiments where phosphorus was added. The growth of the large diatom *Cerataulina pelagica* and the relatively small diatom *Leptocylindrus danicus* was observed with the simultaneous addition of nitrogen and phosphorus, while the contribution of nitrogen was noticeably higher. In an experiment on 11 June 2021, the intensive growth of the large diatom *Proboscia alata* was obtained with a nitrogen addition.

**Table 6.** The results of the factorial experiments on the influence of nitrogen (N) and phosphorus (P) additions on the growth of phytoplankton biomass (mg/m$^3$) in the period 2019–2021. Sxt$_{0.95}$—confidence interval for 5% level of significance.

| Date | Phytoplankton Species | 1 | | 2 | | 3 | | 4 | | Sxt$_{0.95}$ |
|---|---|---|---|---|---|---|---|---|---|---|
| | | N | P | N | P | N | P | N | P | |
| | | − | − | + | − | − | + | + | + | |
| | | 2019 | | | | | | | | |
| 22 May | *Emiliania huxleyi* | 1902.7 | | 1737.5 | | 2154.1 | | 22653.3 | | 551.0 |
| 22 May | *Pseudo-nitzschia delicatissima* | 172.1 | | 1473.1 | | 255.5 | | 6919.3 | | 419.4 |
| 22 May | *Dactyliosolen fragilissimus* | 1464.0 | | 2352.0 | | 1142.6 | | 1496.2 | | 479.8 |
| | | 2020 | | | | | | | | |
| 11 June | *Emiliania huxleyi* | 165.2 | | 168.8 | | 922.6 | | 721.8 | | 78.6 |
| 11 June | *Cerataulina pelagica* | 282.9 | | 2879.9 | | 1767.1 | | 8584.1 | | 1783.2 |
| 11 June | *Leptocylindrus danicus* | 1029.9 | | 2005.9 | | 1632.7 | | 17728.3 | | 903.7 |
| | | 2021 | | | | | | | | |
| 11 June | *Proboscia alata* | 1822.4 | | 4847.4 | | 2325.0 | | 2421.2 | | 788.8 |

**Table 7.** Regression equations showing the effect of nitrogen and phosphorus additives on the biomass of phytoplankton species in the stationary phase of the accumulative culture. Sxt$_{0.95}$—confidence interval for 5% level of significance.

| Date | Species | Regression Equations mg m$^{-3}$ | Sxt$_{0.95}$ |
|---|---|---|---|
| | | **2019** | |
| 21 May | *Emiliania huxleyi* | 7111.9 + 5083.5 N + 5291.8 P + 5166.1 NP | 551.0 |
| 21 May | *Pseudo-nitzschia delicatissima* | 2205.0 + 1991.2 N + 1382.4 P + 1340.7 NP | 419.4 |
| 21 May | *Dactyliosolen fragilissimus* | 1613.7 + 310.4 N − 294.3 P − 133.6 NP | 479.8 |
| | | 2020 | |
| 11 June | *Emiliania huxleyi* | 494.6 − 49.3 N + 327.6 P − 51.1 NP | 78.6 |
| 11 June | *Cerataulina pelagica* | 3378.5 + 2353.5 N + 1797.1 P + 1055.0 NP | 1783.2 |
| 11 June | *Leptocylindrus danicus* | 5599.2 + 4267.9 N + 4081.3 P + 3779.9 NP | 903.7 |
| | | 2021 | |
| 11 June | *Proboscia alata* | 2854.0 + 780.3 N − 480.9 P − 732.2 NP | 788.8 |

## 4. Discussion

Five-year observations have shown that coccolithophores dominate annually in early summer and large diatoms in summer. This trend is consistent with the results of previous studies, which show a similar phenomenon from 2002 to 2017 [68]. Over the past five years, no significant changes in environmental conditions could lead to noticeable shifts in phytoplankton composition and dominant species. The annual dominance of coccolithophores and large diatoms in a specific season is part of the seasonal succession of phytoplankton, the primary regulator of which is the physics of the water column [78,79].

### 4.1. The Bloom of the Coccolithophore Emiliania huxleyi

Two factors determine the water column's hydrophysical processes: the heat flux's intensity at the atmosphere's boundary and the water surface and the wind speed. The wind direction determines the air temperature; NE and SE winds dominate the northeastern part of the Black Sea [80]. The wind speed is maximal in winter and minimal in summer [81,82]. In winter, cold NE winds dominate, the air temperature is below the water temperature, the water's surface is cooled, and convective mixing develops. A thermocline develops with the weakening of the winds and the strengthening of thermal radiation. In late spring and early

summer, with the dominance of weak SE winds, it becomes sharp, and the thermocline depth at this time is minimal. The importance of this parameter for separating the ecological niches of diatoms and coccolithophorids has become apparent only recently [83]. In the Atlantic Ocean, coccolithophore blooms are also observed with the stability of the water column and the presence of a thermocline [84]. High heat flux in late May and early June contributes to increased temperature in the UML, which is close to optimal for the intensive growth of coccolithophores [85,86]. Thus, a sharp gradient thermocline and a temperature close to optimal are necessary to develop *Emiliania huxleyi* blooms.

The intensity of the *Emiliania huxleyi* bloom, expressed as the maximum cells' abundance at the research stations, varied from 2017 to 2021; the maximum was recorded in 2017. The reasons for the interannual variability remain unclear. However, there is a hypothesis that the intensity of the bloom is associated with the intensity of winter mixing; after cold winters with increased vertical turbulence, the bloom is more intense [87]. Indeed, deeper mixing in cold winters transfers denser waters to the upper layers, which enhances vertical exchange when convective mixing stops [88].

The salinity of the UML varies depending on the season, and river runoff plays the primary role [89]. Increased river discharge in spring and early summer leads to minimal salinity. High thermal radiation on the water surface and low inflow of fresh water contribute to some increase in water salinity in summer. However, such insignificant changes in salinity cannot have physiological consequences for *Emiliania huxleyi*, which grows intensively in a wide salinity corridor [90].

The coccolithophore *Emiliania huxleyi* grows intensively and reaches the bloom level at relatively low nitrogen and high phosphorus concentrations. This fact is consistent with earlier data obtained in the Black Sea [67,68]. Furthermore, low nitrogen concentrations are a characteristic feature of blooms in the ocean [90], indicating a similar mechanism of adaptation to low nitrogen concentrations. Furthermore, the half-saturation constant for nitrogen uptake in this species is very low ($\leq 0.5$ μm); this allows it to compete successfully with other species, particularly with the rapidly growing small diatoms *Skeletonema costatum* [91]. The ability of *E. huxleyi* to grow at low nitrogen concentrations [92] and win the competition from other species has an implementation for the "affinity growth strategy" [68]. Indeed, according to the R-competition theory [46,47], the competition will be won by a species with a low residual concentration of the limiting resource.

This strategy implies a low nitrogen quota in the cell [93]. However, the small cell size and the lack of space for vacuoles do not create an opportunity to accumulate intracellular nitrogen reserves. Thus, concerning nitrogen, the *Emiliania huxleyi* cell should be considered not a "storage cell" but an "assembly machinery" according to Klausmeier's classification with co-authors [28,48].

At the same time, *Emiliania huxleyi* demands a high phosphorus concentration; its advantage is manifested at relatively high phosphorus concentrations. Our experimental studies show that it is always necessary to add phosphorus to increase the biomass of this species. Cells of this species grow successfully at N:P below the Redfield ratio. This fact was noted in field studies [68,94] and in experiments with the addition of nitrogen and phosphorus [67]. Increased phosphorus concentrations in the medium suggest an increase in the content of phosphates in the cell. This event usually occurs due to an increase in the content of ribosomes, and according to the hypothesis of the growth rate, this species should be considered rapidly growing. However, in reality, the specific growth rate of *Emiliania huxleyi* is significantly lower than the specific rate of small diatoms. Therefore, this species is classified as having a K-strategy compared to diatoms, which are mostly R-strategists [7]. So far, the reasons for the high phosphorus demands are unclear. Nevertheless, photosystem II activity in *E. huxleyi* decreases faster with phosphorus restriction than with nitrogen restriction, and the opposite happens in diatoms [95]. Also, when phosphorus is limited, morphological changes in both cells and coccoliths are recorded, and the carbon quota in the cell and the degree of its calcification increase [96–98].

Small diatoms are more demanding of nitrogen concentration. In our experiments, adding nitrogen increases their intensive growth (Table 6). In these experiments, N:P is close to the Redfield ratio.

At the beginning of June 2018, in field studies, we found a significant increase in the biomass of the small diatom *Pseudo-nitzschia delicatissima* with an increase in the nitrogen concentration in the medium (CC = 0.48; $p$ = 0.04) (Figure 4). Thus, an increase in nitrogen concentration at a relatively high phosphorus concentration leads to a shift in seasonal succession towards the spring bloom of small diatoms.

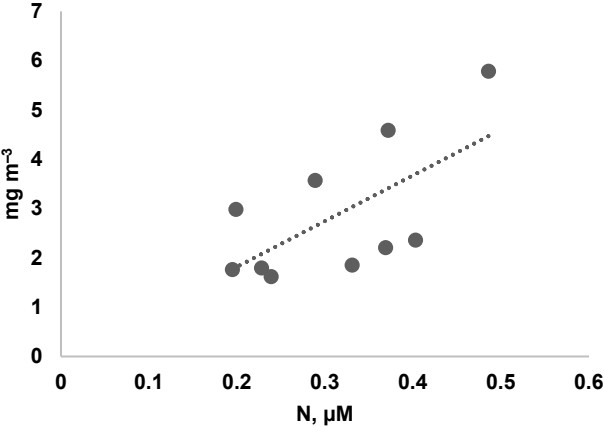

**Figure 4.** Dependence of *Pseudo-nitzschia delicatissima* biomass on nitrogen concentration on 7 June 2018.

### 4.2. Large Diatoms Bloom and the Biogeochemistry of Seawater

Diatoms are the most widespread taxonomic group in the World Ocean, providing a high primary production level and high diversity [8,99,100]. The annual summer dominance of large diatoms in the northeastern part of the Black Sea is associated with the intensive growth of mainly one species, *Pseudosolenia calcar-avis*; sometimes, *Proboscia alata* was recorded.

In the second half of June, the NE wind becomes dominant; its characteristic feature is temporary instability. This event leads to a gradual deepening of the UML, and it ceases to be acutely gradient. The depth of the seasonal thermocline layer increases, and gradients become less steep.

The phosphorus concentration is low and, in some cases, close to zero. An increase in the nitrogen concentration and a significant decrease in the phosphorus concentration lead to an increase in N:P, significantly higher than the Redfield ratio. Due to the intensive consumption of silicon, its concentration is significantly lower than in the bloom period of coccolithophores; the consequence is a decrease in Si:N.

The nutrient ratio and, above all, N:P are regulated by the physics of the water column. The nitrogen concentration increases faster with depth than the phosphorus concentration (see Table S4); at the lower boundary of the thermocline, the nitrogen concentration can reach 2 μM. A noticeable increase in the phosphorus concentration begins from a depth of 90 m. During the summer bloom of large diatoms, mixing reaches depths of elevated nitrogen concentrations but does not reach depths of elevated phosphorus concentrations.

Our experimental results show that the stimulation of the growth of large diatoms occurs with the addition of nitrogen only (Table 5), i.e., with a shift of N:P above the Redfield ratio.

The high N:P in the medium determines the structural stoichiometry, i.e., the high ratio of nitrogen contents in the cell [101,102]. Based on the hypothesis that there is an optimal N:P for each species [28,48], the high N:P we found is optimal for large diatoms. There is a physiological justification for this. The ribosome content in a large cell is relatively low, and the content of vacuoles capable of accumulating nitrogen [103] is high. Concerning nitrogen, large diatoms are storage machines.

### 4.3. Basic Conceptual Scheme

Conceptual schemes reflecting the position of these groups in the coordinates of the physics and chemistry of the water column contribute to understanding the significant differences in the ecological niches of the main taxonomic groups of phytoplankton. The classical scheme is Margalef's Mandala, where turbulence is reflected on one axis and nutrient concentration on the other [104,105]. These schemes have been further developed [106–108].

However, these schemes are of little use for our case because either the nutrients are not separated in them and are considered an integral variable, or the diatoms are considered as a single taxonomic group, without dividing it into dimensional or taxonomic classes. Therefore, in our work, the effect of stoichiometry is reflected in the coordinates N and P. The obtained results allow us to create a unified picture of the influence of nitrogen and phosphorus concentrations and their ratios on the composition of phytoplankton (Figure 4). Low nitrogen concentrations, high phosphorus concentrations, and correspondingly low N:P are the conditions for coccolithophores dominating the community. The dominance passes to large diatoms at high nitrogen and low phosphorus concentrations. Small diatoms win the competition with N:P close to the Redfield ratio and relatively high nitrogen and phosphorus concentrations. Our experimental studies show that almost always, with the simultaneous addition of nitrogen and phosphorus at a ratio close to the Redfield ratio, small diatoms dominate. It is either the pennate diatom *Pseudo-nitzschia delicatissima*, or the small centric *Skeletonema costatum*. Field data demonstrate the dominance of small diatoms during the spring phytoplankton bloom [68]. This event happens in late winter and early spring after the convective mixing stops.

The concentration of nitrogen and phosphorus is high, which allows them to grow at a rate close to the maximum. Under these conditions, species with high maximum specific growth rates benefit. In other words, a rapid growth strategy is implemented in this case [68]. The maximum specific growth rate is characteristic of small diatoms due to the high specific light absorption coefficients, which are several times higher than the same indicator for large diatoms [109]. Comparing Figure 5 we obtained with Figure 1 in the article by Meunier et al. (2017), it should be noted that there is a significant difference. In the segment reflecting the ability of cells to demonstrate P-affinity in Figure 5, there are large diatoms, while in the article by Meunier et al. (2017) [37], there are small cells. Our data demonstrate the high ability of large diatoms to grow at phosphorus concentrations close to analytical zero. In the middle part, we have small diatoms growing at an N:P ratio close to the Redfield ratio [68]; in the article by Meunier et al. (2017), large cells are located in this place. Finally, in our scheme, the species demonstrating high N-affinity is the coccolithophore *Emiliania huxleyi*, instead of the $N_2$-fixation species. Our scheme reflects the results obtained in the Black Sea.

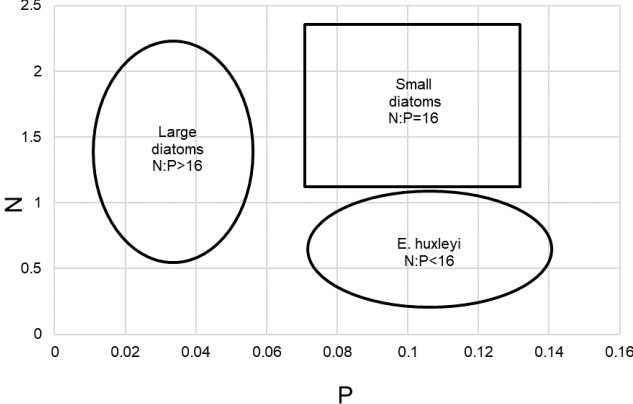

**Figure 5.** Conceptual diagram showing the position of small and large diatoms, as well as coccolithophores, in the coordinates of nitrogen and phosphorus concentrations (μM).

This scheme explains why, with the dominance of coccolithophores, large diatoms are almost absent. With the dominance of large diatoms, the contribution of coccolithophores to the total biomass is negligible. At the same time, the coexistence of species is possible in close niches. Namely, the coexistence of coccolithophorids and *Pseudo-nitzschia delicatissima* is possible. Furthermore, large diatoms can coexist with *Pseudo-nitzschia delicatissima*. These phenomena are shown using our field data.

## 5. Conclusions

Considering the results obtained in the northeastern part of the Black Sea from 2017 to 2021, it should be noted that the ecological succession follows this scheme: small diatoms (spring) → coccolithophores (late spring, early summer) → large diatoms (summer). Furthermore, this scheme was also valid from 2002 to 2016 [68]; thus, there have been no significant changes over the past five years..

Spring phytoplankton blooms occur both in coastal areas and in the open ocean, and they are caused by the intensive growth of small diatoms [106,110,111]. The primary mechanism triggering the spring phytoplankton bloom is based on the hypothesis of the critical depth of Sverdrup and in its various modifications [79,112–114]. In other words, light plays the primary role here as an energy factor. Our research shows that the spring diatom bloom is characterized by the corresponding biogeochemistry, i.e., the ratios of elements close to the Redfield ratio.

The coccolithophore bloom is a widespread phenomenon in the World Ocean (2002), and there are many different hypotheses explaining the mechanisms of bloom development [115] and the role of coccoliths in the prosperity of coccolithophores [89,116]. In this case, the chemistry of the carbonate system plays a unique role [117,118]. Our field and experimental studies show that biogeochemistry is crucial for coccolithophore blooms: they develop at low nitrogen concentrations and N:P below the Redfield ratio. The concentration of silicon does not play any role.

Large diatoms dominate in the community at low phosphorus concentrations and at N:P significantly higher than the Redfield ratio. At the same time, the bloom of large diatoms takes a long time, up to several months. Large diatoms are also successful competitors in the ocean, forming zones of intensive growth from the subtropics to polar regions [117–120]. However, according to generally accepted concepts, the nutrient absorption half-saturation constant is determined by the ratio of cell surface to volume [117,121], and consequently, large diatoms are poor competitors for nutrition. Our studies show that the large diatom *Pseudosolenia calcar-avis* successfully competes for phosphorus and can grow at phosphorous concentrations close to zero.

**Supplementary Materials:** The following supporting information can be downloaded at: https://www.mdpi.com/article/10.3390/jmse11061196/s1. Figure S1. Annual irradiance dynamics on the water surface based on long-term (2006–2021) data; Figure S2. Annual dynamics of water surface temperature based on long-term (2006–2021) data; Figure S3. Seasonal dynamic of water temperature at station with depth 500 m from 2017 to 2021; Table S1. The scheme of experiment for study the influence of nitrates and phosphates supply on phytoplankton growth; Table S2. The phytoplankton composition of the upper mixed layer during the dominance of coccolithophorids in late spring and early summer 2017, 2018 and 2019; Table S3. The composition of phytoplankton of the upper mixed layer during the dominance of large diatoms in the summer of 2017–2021; Table S4. Vertical distribution of nitrogen and phosphorus concentrations during the dominance of coccolithophore *Emiliania huxleyi* (11 June 2020) and large diatoms *Pseudosolenia calcar-avis* (26 August 2021) at the station above a depth of 500 m.

**Author Contributions:** V.S. conducted conceptualization and writing of the paper; L.P. provided the field data and writing of the paper; O.P. provided the hydrophysical data; V.C. provided the hydrochemical data; A.L. conducted experimental studies; A.F. conducted field studies; A.K. data curation. All authors have read and agreed to the published version of the manuscript.

**Funding:** This project received funding from the Ministry of Science and Higher Education of the Russian Federation (theme FMWE-2021–0013) and the Russian Science Foundation grant (project No. 22-17-00066).

**Institutional Review Board Statement:** Not applicable.

**Informed Consent Statement:** Not applicable.

**Data Availability Statement:** The original contributions presented in the study were included in the article and supplementary material; further inquiries can be directed to the corresponding author.

**Acknowledgments:** We appreciate researchers and students who assisted with sampling data.

**Conflicts of Interest:** The authors declare that they have no known competing financial interests or personal relationships that could have appeared to influence the work reported in this paper.

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
