# Peer review of "Phytoplankton Dynamics and Biogeochemistry of the Black Sea"

_jmse, doi:10.3390/jmse11061196_

Round 1

Reviewer 1 Report

The content of the ms is interesting. However, poor data presentation and analysis makes it difficult to understand the ms. Authors have used field level and laboratory scale datasets in the current study, but the results of selective parameters are presented without any justification on why they used so. The results of experiments based on factorial design are not presented in the ms. Overall presentation is confusing and needs severe editing.

Samples collected during numerous cruises from 2017 – 2021 in NE part of Black Sea are used in the current study. Details of each cruise, including the number of cruises in each year and seasons covered may be given.

More details on the factorial design used in the current study may be given. What was the parameters monitored in flasks at different combinations of nutrients are not clear in the methodology.

What was the total phytoplankton abundance in the water column during the study area. Did authors observe the bloom in the entire study area or was it restricted to selected locations? The results related to these aspects in the figure 2 and 3 are confusing.

Author Response

RESPONSES TO REVIEWER 1

SUMMARY RESPONSE: The authors thank the reviewer for detailed analysis of our manuscript. We agree with all the comments, we have tried to take into account all the comments and correct our text in accordance with them.

Reviewer 2 Report

This is an interesting paper.

p.1, 7-8 lines from bottom. Rephrase:  ‘and it accounts for about half of global primary productivity, i.e. carbon dioxide assimilation [1]’

p.1, last 2 lines. The Revelle Factor (CO2 released per CaCO3 precipitated) is less than 1 in seawater (Figure 2 of Frankignoule and Caron 1994 Limnology and Oceanography 39: 458-462).

p.2, State the Redfield Ratio, i.e. an atomic ratio of C106:N16:P1.

p.2, line 7. There is evidence for microalgae that the Growth Rate Hypothesis does not always apply (Flynn et al. 2010 Journal of Phycology 46: 1-12; Garcia et al. 2018 Frontiers in Microbiology 9: 543).

p.2, 9 lines up. Is ‘It is fundamentally essential that’ needed?

p.3, line 12. The Black Sea is not an inland sea in the sense of the Caspian Sea. Like the Baltic Sea, it has a net liquid water flux to the rest of the world Ocean.

p.3, lines 1-2. State that the Redfield Ratio was derived from the C:N:P of phytoplankton averaged over the world ocean and several years.

p.3, lines 3-4. Reference needed.

p.3, line 6. What is meant by ‘structure’.

p.4, line 3; adjust sub- and super-scripts. NO3- NO2- NH4+.

p.4, last 2 lines. Presumably motility could not be measured in formalin-treated samples.

p.5, line 2. Presumably the equation under-estimates fresh weight  in mineralised cells, apart from large diatoms with a large fraction of the cell volume occupied by a low-density vacuole.

p.5, line 8. ‘filtered’, not ‘filtrated’. Also, what about unicellular zooplankton?

p.5, line 10. Clarify ‘uually’.

p.5, line 19. Picoplankton is usually defined as 0.2 – 2 μm.

p.8, lines 4-5. Clarify ‘layer below the thermocline’.

p.8, line above Table 4. Does ‘not reliable’ mean ‘variable’? Also, what organisms comprise the bloom?

p.9, Table 5. ‘Mg’ should be ‘mg’.

p.10, line 8. ‘blooms’, not ‘bloom’.

p.10, line 9. ‘The intensity of the bloom in the years of the research was variable’. Also, what organism constituted the blooms?

p.10, 17 lines up ‘acclimation’ (phenotypic), not ‘adaptation’ (genotypic’).

p.10, 8 lines up. Clarify ‘storage cell’ and ‘assembly machinery’.

p.11, line 2. ‘r’, not ‘R’.

p,11, 6 lines up. ‘and gradients become less steep’.

p.12, line 2.  Should ‘Figure S’ be ‘Figure 5’.

p.12, lines 10-11. Reference needed for the ribosome content of the large cells, and whether the ribosome content is on a whole cell basis or a cell minus vacuole basis. Also, what is the relevance of the protein content of chloplasts?

p.12, lines 17-19. Good point.

p.13, figure 5, Clarify what the 3 axes other than those indicating N and P concentration refer to?

p.14, line 5. Explain ‘flowering’.

Needs a little correction.

Author Response

RESPONSES TO REVIEWER 2

SUMMARY RESPONSE: The authors thank the reviewer for a thorough and detailed analysis of the manuscript. We agree with all the comments, we took into account all the recommendations and we have made the appropriate corrections in the text.
